# Feedbacks on a Machine Learning Curriculum for an International Audience

## Abstract

Teaching Machine Learning (ML) can be challenging due to the breadth of the subject and the diversity of audience that might be interested in it. In this article, we collectively place ourselves both at the curriculum level and at the lesson level, describing existing practices and sketching directions for improvement. We first describe a curriculum for ML practitioners that involves different constraints including the variety of audience, both in terms of background and in terms of learning goals. We also explain how some of the lessons relate to the teaching principles that are pushed forward for example by *The Carpentries* and how the future of the curriculum could be reshaped.

## 1. Introduction

With the maturation of the field, learning fundamentals of Machine Learning (**ML**) can become a key differentiator for learners and researchers of all domains. Indeed, many application domains and research domains are being structurally transformed by the availability of data and the use of ML. In this context, effectively *teaching ML* becomes of utmost importance so that ML practitioners understand the tools they manipulate and are able to creatively and robustly use them. Teaching ML, however, presents many challenges due to this pervasive applicability.

Probably one of the biggest challenges is the heterogeneity of the possible audience. This compounds with the fact that ML is at the crossroads of computer science, maths and statistics. As in the traditional lessons from *The Carpentries*, a part of the target audience might have none or very limited programming background. The same goes for mathematical background, where math anxiety is a common issue across disciplines. As far as statistics are concerned, almost all scientists would benefit from a better statistical background. In Section 2, presenting an existing curriculum,

we describe the target audience(s) we consider and how we have mitigated for the differences in background.

Another challenge is the one of deciding of what topics should be taught. While a mostly practical approach can be tempting, having only a shallow understanding of the ML methods often leads to unsatisfying results, waste of time and incapacity to innovate. Indeed, in addition to practice, the learners must be able to analyze and understand why, when, and how the methods they use work. In Section 2, we additionally present our target goals and guidelines in the curriculum. We concisely focus, in Section 3, on some illustrative practice in our curriculum, and group in Section 4 the results of our reflection on the evolutions of the curriculum.

## 2. A Modular Machine Learning Curriculum

In this section, we sketch an existing ML curriculum that has a 3-semester duration, in a university context. We focus on the core curriculum but also present how part of it is addressed and fined tuned to different audiences with different goals. While it surely can be improved a lot, this curriculum can be used as a basis for reasoning about the topics of interests in an ML curriculum.

### 2.1. Audience

The core curriculum has ML at its core (together with symbolic Artificial Intelligence and Data Mining). While examining hundreds of applications every year, it is more and more common to see applicants that have already followed some introductory MOOC on Machine Learning. Despite this, while learners are supposed to be at ease with programming, their actual programming level shows a great discrepancy. Similarly, an introductory-level statistical background is presupposed but in practice, its mastery is variable.

The audience is also international, with people coming from tens of different countries; all lessons are in English, which is not the mother tongue of most of the audience, adding an additional difficulty to the understanding. This difficulty is worth its benefits. The diversity of linguistic, cultural and work experience[1] backgrounds makes the learning experience richer and more open on many aspects.

[1] Anonymous Institution, Anonymous City, Anonymous Region, Anonymous Country. Correspondence to: Anonymous Author <anon.email@domain.com>.

Preliminary work. Under review by the Teaching Machine Learning Workshop at ECML 2021. Do not distribute.

[1] Applicants range from students to people having worked for a few years in various domains to people coming back to studies after decades of work in the industry.

**Secondary audience(s).** In addition to the main audience, other audiences reuse the presented lessons. Some lessons are directly shared (mixing learners and thus increasing the benefits of heterogeneity), some are tuned to the audience to cope for background discrepancies. These audiences include learners that aspire to become software engineers with a data science background, but also learners from other disciplines (e.g., Optics, Physics) that want to learn ML.

### 2.2. General Goals and Guiding Principles

We aim to prepare the learners for **ML research and innovation** while acquiring the necessary skills to **solve data-driven problems**. So, the ultimate goal is to provide a strong background on the entire ML pipeline from the raw data to the final ML software and its impact on the application domain.

**Preparing for research and innovation.** To be able to do advances in research or to push the bleeding edge of innovation, a superficial knowledge is insufficient. We aim at making our learners being able to imagine/visualize the data, to understand how each type of model processes these data and to understand the dynamic of the training process of these models. Such deep understanding makes it possible diagnose unexpected problems and limitations of the models in particular situations. Most importantly we believe they are mandatory to be able to improve existing methods, propose novel models, or formalize a new problem to be solved.

**Learning to carry out a data project.** All ML methods are driven by the data and the task at hand. One of the objectives is to make the learners aware that raw data is by definition not smooth and that *analyzing, understanding and pre-processing* the data are mandatory tasks. Given the data (and their characteristics), the learners have to be able to **mathematically formalize the issue** to propose or to design ML solutions. A key point is the ability to perform experiments to validate the methods (and the results) and to design **a relevant and rigorous experimental protocol**.

**Acquiring transversal skills.** At any stage, three cross-cutting aspects are very important for our learners:
• *working in teams* especially when software implementation plays a central role.
• *presenting their work* orally or in writing.
• *understanding the broader impact* of ML design choices, like fairness, privacy or environmental impact.

### 2.3. ML Curriculum: a quick tour

The current embodiment of the curriculum is sketched in Fig. 1. With the increase of applications (thus their diversity) and the maturation of the domain, the curriculum must be continuously evolving. This evolution driven by systematic feedback gathering from learners and teachers, but also from

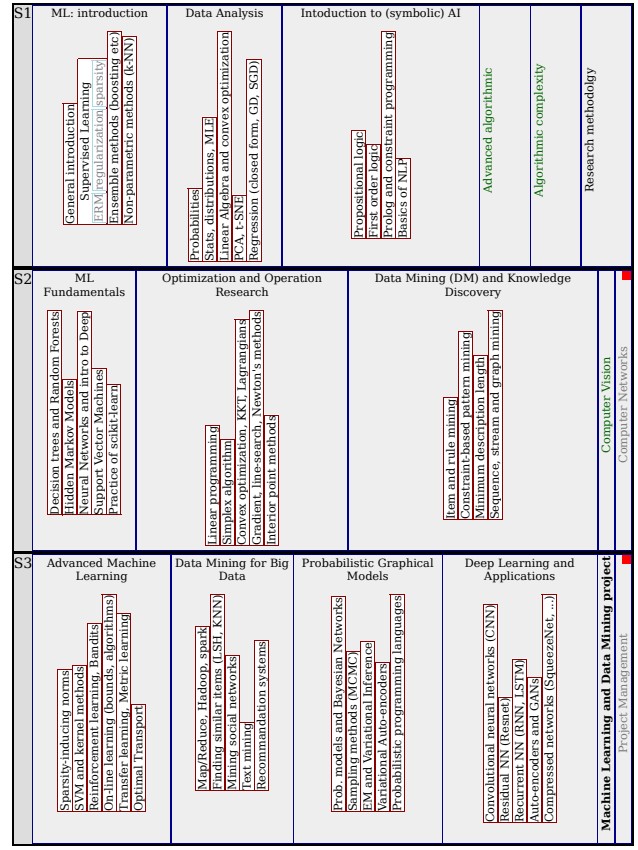

*Figure 1.* Summarizing view of an existing 3-semester university curriculum for Machine Learning, the fourth semester being dedicate to an internship (in a research facility or in a company).

the research labs and companies that hire our learners. The curriculum aims at achieving the presented goals, mainly by covering a broad variety of machine learning topics in great details and by including a lot of projects. Some lessons are more targeted at strengthening the computer science background, while others focus on the transversal skills.

## 3. Selected Lessons and Practices

In this section, we focus on some elements of the current curriculum, starting from the most general ones to the ones that are most specific to Machine Learning (ML).

**Distributed learning.** We intentionally cover some topics repeatedly across the curriculum, being fundamental skills like probabilities, optimization, algebra, or more specific ML models (as SVM). Such an approach helps consolidating the understanding of these concepts. At a finer scale, many lessons spread a subject over a semester, first presenting a theoretical concept that is soon used in practical sessions and later in a project.

**Progressive unscaffolding.** Scaffolding, which lowers the cognitive load by providing learners with a structure for a solution to a problem, has been shown to be effective. We follow a progressive unscaffolding principle in the learners challenges: at first, most of the code is provided, either in the form of jupyter notebooks (with more or less things to complete) or of code to copy and adapt, then, we might ask to implement or study a given algorithm, and finally, we end up with very open projects where learners need to make most of the decision, starting from scratch.

**Heterogeneity and team work.** One specificity of our program is the heterogeneity of the audience which is an interesting playground for developing specific skills. Indeed, in their future professional life, learners will not work alone, or with competent and friendly colleagues whom they know well and on pleasant tasks from scratch. Part of the instruction must therefore prepare learners for the real-life situations they may encounter in their professional lives. In many lessons, learners are thus asked to work in larger or smaller groups, on more or less complex projects, of longer or shorter duration, in more or less complex environments. The interesting skills worked on here are: the ability to carry out several projects at once, to plan each of them, to build an efficient project team with colleagues whom they do not necessarily know and whose level they do not necessarily know at the beginning, etc. Other skills that are more related to interpersonal skills are also developed: the ability to adapt to the working methods of colleagues, to work under pressure, to communicate with others to avoid clashes, etc.

**Problem based learning.** During the curriculum, several practical projects are proposed. An important one is the one proposed in the last semester, in which the learners are participating to a national data challenge proposed by industrials and open to different universities. The topic and the type of data change each year: anomaly detection from sensor data provided by Airbus, temperature prediction and uncertainty estimation, identifying the presence of a wind turbine from satellite images to name a few. These projects last during one semester, and allow learners to discover methods or applied problems that are not necessarily covered during other courses. They are participating in teams (of 2 to 4) and are also encouraged to discuss with other teams The output obtained by learners on a test set can be submitted online at any time during the semester, and the results are displayed and updated on a public leader-board at every new submission. At the end of the challenge, 3 teams from the top 10 are invited for a presentation during a day dedicated to the results summary and open to all participants. We also ask the learners to write a small scientific article explaining how they obtain their results (state-of-the-art, methods tested, chosen solution and experimental details).

**To model.train()... and beyond!.** Most of ML courses aim at making learners able to train a prediction model in a Jupyter notebook (and later submitting jobs to a GPU cluster). By doing this, learners feel the need for early data analysis and data wrangling, and then have a first-hand experience with various ML techniques. There is however a gap between ML a model in a notebook interface and then making it available in a corporate software infrastructure. We therefore include in some of our curriculum wider learner projects in which the learners learn how to glue together the different aspects needed to exploit a model in a distributed software architecture. This usually includes extracting datasets from a Hadoop cluster, data wrangling on the fly and store the processed data in a NoSQL database, and then use their prediction model published as a microservice which will store its output in the NoSQL database as well. We believe that such project let the learners reharsh several aspects of machine learning and data mining oriented software development, as well as giving them the opportunity to think and create complete software architectures and data pipelines which include machine learning. This part of the curriculum is of the utmost practical interest for those who will likely serve as research engineer later.

**Towards responsible ML.** We believe that it is important to teach learners about the fact that they have to think in a responsible manner on the societal impacts of what they will do as future scientists. To do so we have teach them how to be transparent and honest by designing reproducible methods (building proper pipelines). This honesty is achieved in part by learning ML models properly: Building an effective ML model relies heavily on selecting its parameters, and teaching the correct way to validate them is a key part of the ML pipeline (e.g., cross-validation). Moreover, the learners need to be aware that biases in ML can have multiple sources: the data obviously but also the model at hand. Those biases reflect our human perception of the world and our societies, therefore they are likely not totally avoidable, but being able to study the societal impacts of ML methods is key. Raising these issues in the curriculum should not be an option, and should be taught as a common thread within the various courses, through explicability, fairness, privacy, environmental impact, etc.

## 4. Moving Forward with the Curriculum

**From toy dataset to real-life/industrial dataset.** To learn the principles of ML methods, toy datasets (e.g., 2D datasets, MNIST[2] or UCI[3] datasets) are often used. On the one hand, with this kind of data it is "easy" to visualize and understand the basic behavior of the methods. On the other hand, this kind of data is "too simple/too clean" to be aware of the need for a thorough study of the data to understand its

---

[2] http://yann.lecun.com/exdb/mnist/
[3] http://archive.ics.uci.edu/ml

characteristics and to have the intuition of which method(s) to use (either in pre-processing the data or for the learning phase itself). Indeed, when the learner is confronted with real/industrial data, he/she will be faced with different types of problems. Among these, we can mention noise and/or outliers, the amount of data available, unbalanced data, distribution drift, etc. These notions are explained during the curriculum and can be applied by the learners in particular during the challenge proposed at the end of the curriculum, we think that this only allows to glimpse the tip of the iceberg. Obviously, in an academic training context it is not possible to train learners on all the issues they may face during their career. However, it would be appropriate to offer projects with increasingly complex data during the curriculum, and to adapt the follow-up so that learners can have a discussion with the teachers more frequently in order to highlight the problems they face and discuss the solutions implemented.

**Faster learner/teacher feedback loops.** Currently, learners mostly receive feedback at the end of semesters, with final exams and project defenses. Similarly, the teacher only have feedback at the end of each semester through a (long) form that each learner have to fill out, the heterogeneity of the audience clearly makes such form not enough informative. We can imagine to tighten this feedback loop by having periodic discussions between the learners and teachers in a similar way to what is done in software development with agile methods and stand up meetings every Monday morning. This method would have the advantage to *(i)* allow teachers to quickly adapt the course regarding the heterogeneity of the learners and *(ii)* to better catch up on a course where the learners can have difficulties.

**Enforcing prerequisites, embracing diversity.** An approach to align the curriculum and the learner population is to enforce prerequisites regarding academic background. Providing a list of prerequisites so that learners can self-assess their background is often too imprecise to be sufficient. One solution, which is very costly in human resources, is to require applicants to solve and hand in a project and also to conduct interviews. A promising intermediate solution that we would like to explore consists in suggesting online resources to acquire the necessary prerequisites, and providing an automated platform on which applicants should validate exercises.

Convenient as it may be, enforcing strong uniformity is not in the spirit of our international program. We strongly believe that systemic diversity is desirable: it develops learners openness to different cultures, encourages peer based learning and promotes group cohesion.

**Generalizing helping visualizations.** "A picture is worth a thousand words" and an interactive animation is worth thousands pictures. As a illustration of this, the addition of many pictures to the "Version Control with Git" lessons from Carpentry[4], pushed this course to a new level and, in our experience, made it a lot more accessible for newcomers. Beyond providing sound mathematical formulations, we want to actively increase[5] the amount of visualizations we provide to our learners. With the advance of technologies like dynamic websites and notebooks, one can even create interactive illustrations, i.e., playgrounds, in which the learners can play with a model to better understand how it reacts to various inputs and/or parameters. In addition to pointing our learners towards broadly known interactive animations, e.g., for neural nets (MLP[6], ConvNets[7][8][9]) or the very well polished Distill[10] platform, we are committed to continuously creating new visualizations and interactive codes that favor deeper understanding.

**Carpentries-style homogeneous lesson format.** While our current curriculum is roughly split into lessons (red boxes in Fig. 1), uniformizing the format/template and explicitly structuring the lessons using the notion of episodes (an existing example is given in light grey in Fig. 1) could be interesting. Indeed, having the equivalent of the Carpentries' workshop *schedule page* and the *reference pages* is very useful, and could advantageously replace the course description sheets. This would also allow to recombine the lessons to more easily create new curriculum for different audiences. An important caveat is to carry this uniformization while allowing for originality and easy experimentation of new content or practice.

**An condensed ML-carpentry for all researchers.** While, our curriculum is aimed at giving learners strong ML foundations over the course of 2 years at the master level, we aim at reusing lessons and episodes. We would like to author and propose a doctoral level courses on ML for Ph.D. students and researchers of different domains. The format could be very close to the workshops from The Carpentries, but spread over a few weeks to leverage distributed practice. More modest learning goals are targeted: understanding ML fundamentals, practicing one framework and understanding deeply at least one method. Several follow-up "specializations" could be proposed, e.g., some on specific approaches (e.g. deep learning) and one focusing on the data pipeline.

## 5. Conclusion

We presented an existing ML curriculum, highlighting some features and possible improvements. We hope this presentation can help fueling discussion around Teaching ML.

---

[4] https://swcarpentry.github.io/git-novice/

[5] To preserve anonymity, none of our visualizations are used as examples.

[6] https://playground.tensorflow.org/

[7] https://convnetplayground.fastforwardlabs.com

[8] https://poloclub.github.io/cnn-explainer/

[9] https://www.cs.ryerson.ca/~aharley/vis/conv/

[10] https://distill.pub