# OpenReview forum: "Feedbacks on a Machine Learning Curriculum for an International Audience"
_ecmlpkdd.org/ECMLPKDD/2021/Workshop/TeachML — Submitted to TeachML 2021_

### Official Review · Reviewer_NVHr · 2021-07-15
**Interesting paper about a modular ML curriculum with some writing flaws**

**Rating:** 6
**Confidence:** 4

**Review:**

This paper is about a modular curriculum for teaching machine learning for international students, over three semesters. The curriculum seems to be well designed, with clear goals, guiding principles, and lots of discussion on students' learning difficulties and future changes to curricula.

It is very positive that the paper defines the audience of the course, with clearly defined target populations, and even a secondary audience. This is also supported by the modular curriculum as it can adapt to diverse audiences.

I think this paper could lead to interesting discussion in the workshop. The paper adopts many of the Carpenties framework for its teaching methodologies.

Some minor issues:

- The paper does not have references, which I find strange, only some links provided in footnotes, these should be full references in a well made references or bibliography section. No paper is disconnected from the literature. For example the Carpentries manual should be cited, or papers about progressive scaffolding, etc.
- I am not convinced on the title, in particular the word "Feedbacks", not sure what it means here, sounds like the title is asking for feedback, but the title should describe the paper contents, not what other people might do with the paper. I suggest the authors to think on a more descriptive paper title, I would use the words Modular ML curriculum as this is an interesting feature of this paper.
- Figure 1 is a bit hard to read due to the rotated text, but I think this is due to limited space in the paper.

---

### Official Review · Reviewer_9XTG · 2021-07-16
**Report on a promising ML curriculum, however lacking expositoin and/or quantifiable feedback**

**Rating:** 4
**Confidence:** 3

**Review:**

The paper introduces a ML curriculum to teach a diverse audience machine learning techiques in the course of a 3-semester span within a university content. In the paper a number of  goals are explained (Sec 2.2) and overarching techiques  are listed (Sec 3.) In Section 4 a number of topics are touched on that would further improve the currriculum.

However, generally the paper lacks detail. Crucially the actual curriculum is not explained in the text beyond Fig. 1 and only little specific information is provided as to the motivations of why the curriculum is structured as it is. Section 2.3 ("ML Curriculum: a quick tour") is merely 11 lines of text whereas I would have expected it to be a major section in the paper. The techinques listed in Section 3 remain vague and few or no examples are given how they are concretely applied in the curriculum, e.g. "Distributed learning"  only mentioned SVMs, no concrete example is given where progressive unscaffolding has been applied or where, concretely, in the curriculum team work is required. In "Problem-based learning" it is uncear why it is advantageous to change the data used in the curriculum every year. While the importance of "responsible ML" is stressed,  little explanation is given as to how the authors aim to convey it.

The paper also has almost no references even though a number of claims are made that could use them. In the introduction it's is said that "math anxietty is a common issue" or "almost all scientists would benefit from a better statistical background" which either has scholarly work to back it up with or the text should be revised to be less authoritative.

Lastly, the paper would greatly benefit from empirical data. In section 4 feedback loops are discussed and teacher and student evaluations are mentioned suggesting a interesting source of data, that the reader of a paper on "Feedbacks .. on a Curriculum" would be interested in. How was the curriculum rated, did the students effectively learn?

The curriculum generally sounds interesting,  a improved version of the paper would be a valuable contribution.

---

### Decision · Program_Chairs · 2021-07-21

**Decision:**

Reject

**Comment:**

Thank you for submitting this year to the Teaching ML workshop. The reviewers agree that this paper is not ready for publication.

We encourage the authors to keep up their efforts in the field and act upon the suggestions made. We would love to see a submission from you in the future. We cordially invite you dial in for the workshop itself to be part of our community and make contributions there. We are looking forward to hearing from you in the future.